# Differing Effects of Body Size on Circulating Lipid Concentrations and Hemoglobin A1c Levels in Young and Middle-Aged Japanese Women

**DOI:** 10.3390/healthcare12040465

**Published:** 2024-02-13

**Authors:** Katsumi Iizuka, Kazuko Kobae, Kotone Yanagi, Yoshiko Yamada, Kanako Deguchi, Chihiro Ushiroda, Yusuke Seino, Atsushi Suzuki, Eiichi Saitoh, Hiroyuki Naruse

**Affiliations:** 1Department of Clinical Nutrition, Fujita Health University, Toyoake 470-1192, Japan; kanasakuran@gmail.com (K.D.); chihiro.ushiroda@fujita-hu.ac.jp (C.U.); 2Food and Nutrition Service Department, Fujita Health University Hospital, Toyoake 470-1192, Japan; 3Health Management Center, Fujita Health University, Toyoake 470-1192, Japan; kobae@fujita-hu.ac.jp (K.K.); yanagi-k@fujita-hu.ac.jp (K.Y.); yoshiko.yamada@fujita-hu.ac.jp (Y.Y.); hnaruse@fujita-hu.ac.jp (H.N.); 4Department of Endocrinology, Diabetes, Metabolism, Fujita Health University, Toyoake 470-1192, Japan; seinoy@fujita-hu.ac.jp (Y.S.); aslapin@fujita-hu.ac.jp (A.S.); 5Department of Rehabilitation Medicine I, School of Medicine, Fujita Health University, Toyoake 470-1192, Japan; esaitoh1@me.com; 6Department of Medical Laboratory Science, Fujita Health University Graduate School of Health Sciences, Toyoake 470-1192, Japan

**Keywords:** HDL-C, high-density lipoprotein cholesterol, non-HDL-C, non-high-density lipoprotein cholesterol, underweight, normal weight, overweight

## Abstract

The condition of being underweight is a social problem in Japan among women. However, there is a lack of evidence for dietary guidance for underweight women because there has been no comparison of lipids or HbA1c among underweight, normal weight, and overweight women in different age groups. We analyzed the effect of body size and age on the serum lipid and hemoglobin A1c levels in Japanese women in a cross-sectional study. A total of 26,118 women aged >20–65 years underwent physical examinations between 2012 and 2022. Seventeen percent of women aged >20–29 years were underweight, and 8% of those aged 50–65 years were underweight. Total cholesterol and non-HDL-C concentrations increased with age, but the difference between underweight and overweight individuals was lowest among women aged 50–65 years. On the other hand, the differences in HDL-C, TG, and HbA1c levels between underweight and overweight subjects were greatest in the 50–65 age group, but the differences between underweight and normal weight subjects were much smaller. Considering that, unlike HDL-C, TG, and HbA1c, TC and non-HDL-C increase to levels comparable to overweight levels in underweight women in aged 50–65 years, educating people about a diet that lowers non-HDL-C is necessary even in young underweight women.

## 1. Introduction

Young women in Japan have lower body masses than women in Western countries [1,2,3,4,5]. The health risks associated with underweight individuals are generally underestimated in comparison with those associated with overweight individuals in routine practice, especially in preventive medicine. Women are more likely to be underweight than men are, and those who are underweight are at a higher risk of a number of conditions, including osteoporosis, thinning skin, dry skin, hair loss, poor dental health, anemia, immune deficiencies, infertility, premature birth, and early menopause [1]. Moreover, underweight women are at a higher risk of vitamin deficiency, such as vitamin D, vitamin B_12_, and vitamin B_1_, and folate deficiency [2]. Furthermore, being underweight is associated with a greater risk of all-cause mortality than at normal weight [6]. Thus, low body mass in women poses various health risks and appears to be a major health issue in Japan.

Previous studies conducted in Japan have shown no differences between underweight, normal weight, or overweight women regarding their risk of high total cholesterol (TC) concentrations (>220 mg/dL (5.69 mmol/L)) [7]. However, it is unknown whether circulating TC and non-high-density lipoprotein cholesterol (non-HDL-C) concentrations increase with age, even in underweight women. Furthermore, there have been no studies in Japan providing a basis for the type of nutritional guidance that should be provided for such women.

Lipid metabolism differs considerably between men and women [8]. This difference is particularly pronounced in premenopausal women who have low TC, triglyceride (TG), and low-density lipoprotein-cholesterol (LDL-C) concentrations and a high HDL-C concentration [8]. However, the TC and LDL-C concentrations in postmenopausal women are greater than those in premenopausal women; therefore, their risk of atherosclerosis is greater [8,9,10]. In addition, menopause causes estrogen concentrations to decrease, which is associated with an increase in the risk of atherosclerosis [8,9,10,11,12].

Glucose tolerance deteriorates with age [13,14,15], and age-related glucose intolerance in humans is accompanied by insulin resistance or impaired secretion [13,14,15]. In contrast, although thin people are less likely to develop dyslipidemia and diabetes, studies examining the extent to which differences in body size affect the age-related deterioration in glucose tolerance or what initiates this deterioration are rare.

In the present study, we used 10 years of health examination data to determine the effects of body size on carbohydrate and lipid indices in women in a number of age groups (20–29, 30–39, 40–49, and 50–65 years of age). If the results of the study clarify the relationships between underweight status and carbohydrate and lipid metabolism in women in each age group, they could help educate people regarding diets that can be adopted to prevent atherosclerosis from an early age.

## 2. Materials and Methods

We performed a cross-sectional study using health examination data collected between 2012 and 2022 because although some of the data were collected from the same participants, there was no year-to-year correspondence. We calculated the proportions of women who were underweight, normal weight, or overweight in four age groups: 20–29, 30–39, 40–49, and 50–65 years. Furthermore, the changes in circulating parameters (TC, non-HDL-C, HDL-C, TG, and HbA1c) in each age group were evaluated for each body size group. Finally, we compared the differences in these blood parameters between 2012 and 2022 among the four age groups according to body size. The data were provided by the Fujita Medical University Health Care Centre in a fully anonymized form such that they were depersonalized. These anonymized data were accessed on 14 June 2023.

The data obtained were age (years), body mass index (BMI, kg/m^2^), TC (mg/dL), HDL-C (mg/dL), TG (mg/dL), non-HDL-C (mg/dL), and HbA1c (%). The height and body mass of the participants were measured by a nurse during the examinations, and the lipid and HbA1c levels were measured in the laboratory of the same hospital. The plasma lipid concentrations (TC, TG, and HDL-C) were measured using a Hitachi LABOSPECT008 (Hitachi High-Tech Corporation, Tokyo, Japan), and HbA1c was measured using an A1c HA-8190 (Arkray, Kyoto, Japan). Non-HDL-C concentrations were calculated using the TC and HDL-C concentrations. Blood samples were collected at approximately 16:00–17:00 from non-fasting participants. Since this was a hospital staff medical checkup, the checkup was set for after 3:30 p.m., when medical staff have finished seeing outpatients.

The participants were allocated to groups according to body size (underweight, BMI <18.5 kg/m^2^; normal weight, BMI ≥18.5 and <25 kg/m^2^; or overweight, BMI ≥25 kg/m^2^) and age (20–29, 30–39, 40–49, or 50–65 years). Age, BMI, TC, TG, HDL-C, non-HDL-C, and HbA1c values are presented as the mean ± standard deviation. Analysis of covariance (ANCOVA) was performed to compare each parameter (TC, TG, HDL-C, non-HDL-C, and HbA1c) among BMI groups by age subgroup. The explanatory variables for ANCOVA included year and age as well as BMI group to avoid confounders. *p* < 0.05 was considered to indicate statistical significance. Comparisons between the groups were performed using the Bonferroni correction. Statistical analysis was performed using SPSS version 28.0.0.0 for Mac (IBM Corp., Armonk, NY, USA).

The study was conducted according to the principles of the Declaration of Helsinki and was approved by the Research Ethics Committee of Fujita Health University (approval number HM23-053). Because the study was conducted using anonymized data, it was not possible to obtain the consent of the participants.

## 3. Results

More than 2000 women underwent health checks annually between 2012 and 2022. The mean age of the participants was approximately 34 years, and their mean BMI was approximately 21 kg/m^2^ (Table 1). Forty five percent of the participants (n = 11,840) were aged >20–29 years, 24% (n = 6160) were aged 30–39 years, 19% (n = 5019) were aged 40–49 years, and 12% (n = 3099) were older than 50 years (Table 1). We identified the metabolic indices of the participants who were underweight, normal weight, or overweight in each age group. The proportions of participants in the 20 to 29 year-old group, 30 to 39 year-old group, 40 to 49 year-old group, and 50 to 65 year-old group were 45% (n = 11,840), 24% (n = 6160), 19% (n = 5019), and 12% (n = 3099), respectively. The 20–29-year-old group had the highest proportion of participants who were underweight (n = 2059, 17%) and the proportion decreased across the 30–39 (n = 882, 14%), 40–49 (n = 449, 9%), and 50–65-year-old age groups (n = 251, 8%). However, the 20–29-year-old group had the lowest proportion of participants (5%) and the proportion increased to 18% in the 50–65-year-old age group (Table 1).

We subsequently performed ANCOVA to test the effect of body size and age on plasma lipids and HbA1c levels. Among individuals aged >20–29, 30–39, and 40–49 years, participants who were overweight had the highest TC concentrations, while those with a normal weight had slightly greater TC concentrations than those who were underweight. However, there were no differences in the TC concentration between the participants who were underweight or had a normal weight or between those who were underweight or overweight in the 50–65-year-old age group (Table 2). In fact, the mean TC levels in participants who were underweight, normal weight, or overweight and aged 50–65 years were 218, 220, and 215 mg/dL, respectively.

Similarly, the differences between underweight and overweight group in non-HDL-C levels in the 50 to 65 year-old group were similar to those in total cholesterol. The difference in non-HDL-C between overweight and underweight increased to −27.9, −32.7, and −34.1 in the 20–29, 30–39, and 40–49 age groups, but conversely decreased to −20.3 mg/dL in the 50–65 age group (Table 3). Among the participants aged >20–29 or >30–39 years, the non-HDL-C concentrations were highest in the overweight group and lowest in the underweight group. However, the differences in non-HDL-C concentrations between body-size groups were smaller for participants in the 50–65-year-old group (Table 3). 

As described above, the differences in TC and non-HDL-C levels between the overweight and underweight individuals aged 50–65 years were smaller than those between those aged >20–29, >30–39, and >40–49 years. As non-HDL-C is calculated by subtracting total cholesterol from HDL-C, we tested whether age and body size affected HDL-C. Regarding HDL-C, participants who were overweight and aged >20–29 years and aged 30–39 years had the lowest HDL-C concentrations, whereas those with a normal weight or who were underweight had similar HDL-C concentrations. In contrast, there were markedly greater percentages of underweight individuals aged 50–65 years than of those aged >20–29, 30–39, or 40–49 years (Table 4). Thus, the effects of body size on HDL-C were different from those on TC and non-HDL-C.

TG concentrations were highest in participants who were overweight and were similar in those with a normal weight or who were underweight. Among all the age groups, the difference in TG concentration between the participants who were underweight and those who were overweight was >40 mg/dL (0.45 mmol/L). The differences in TG concentrations between participants who were underweight and those with a normal weight or who were overweight were greatest in the 50- to 65-year-old group (Table 5). TG levels in overweight participants aged 50–65 years reached 150 mg/dL.

In all the age groups, the HbA1c values for participants who were overweight were greater than those for participants in the normal-weight and underweight groups. The HbA1c level in overweight participants aged 50–65 years reached 6.0%. The differences in HbA1c between the participants who were underweight and those who were overweight were greatest in the 50- to 65-year-old group (Table 6). However, there was no difference in the HbA1c values between the participants with a normal weight and those who were underweight (Table 6).

## 4. Discussion

In the present study, we first clarified the effects of low body weight on lipids and HbA1c levels in the world by age in underweight Japanese women. The proportion of women who were underweight decreased with age but remained at 8% for those aged 50–65 years. The differences in the TC and non-HDL-C concentrations between the women who were underweight or overweight and aged 50–65 years were much smaller than those between women aged >20–29 years. In contrast, the differences in TG and HbA1c between underweight and overweight women aged 50–65 years were much greater than those between underweight and overweight women aged >20–29 years. These findings suggested that cholesterol metabolism was likely affected by menopause rather than by body size and that HDL-C, TG, and HDL-C were affected by body size and age. 

Being underweight was found to be much more common than being overweight in women aged >20–39 years, but this was reversed in those aged 40–49 years. These trends were found over a 10-year period. Consistent with the present data, the percentages of Japanese women in their 20s with a BMI <18.5 kg/m^2^ were 13.5% in 1979, 20.4% in 1998, 22.3% in 2009, 21.5% in 2013, 20.7% in 2016, and 20.7% in 2019 [16,17,18,19,20]. An underweight BMI <18.5 kg/m^2^ is associated with lower fertility [21,22], and a low BMI is associated with a lower probability of a woman becoming pregnant, a higher risk of miscarriage, and a lower probability of live birth [23]. A low BMI is also associated with anemia, low sex hormone concentrations, low bone density, hypotension, and feelings of fatigue and malaise [1]. We have also reported that young women who are underweight are predisposed to deficiencies in vitamins B1, B12, and D and folate [2]. This change was also observed in underweight women older than 40 years. Given the health risks of being underweight described above, the fact that the prevalence of underweight in women has not changed (at 20%) over the last 20 years is an issue that urgently needs to be addressed in Japan [16]. Some studies have reported that associations between BMI and mortality are stronger at younger ages than at older ages. Compared with that of individuals with a healthy weight (BMI 18.5–24.9 kg/m^2^), the life expectancy from the age of 40 years was 4.3 and 4.5 years shorter in underweight men and women, respectively [24]. Therefore, our results also suggested that medical policies for young underweight women in Japan are urgently needed.

For women aged 40–65 years, the ratio of overweight to underweight individuals was opposite to that for younger women, but nutritional deficits were present in participants who were either underweight or overweight in the present study. The percentage of 50-year-old women with obesity was as high as 18%. This is analogous first to the change in eating habits due to the change in the environment caused by marriage. This approach is analogous to the fact that most of the population are nurses whose workload decreases as they age due to changes in the nature of their work (which is primarily the management of subordinates). Thus, this implies that different age groups have different challenges related to malnutrition. Since 8% of the population in their 50s are underweight, it is necessary to address both overweight and undernutrition. Some studies have reported that, compared with that of individuals with a normal weight, the life expectancy from the age of 40 years was 3.5 years shorter for obese women [24]. In addition to being underweight, medical policies against obesity are needed for women aged 50–65 years.

In the present study, we showed that non-HDL-C and TC levels increase with age, and the concentrations in women who were underweight and aged 50–65 years were similar to those in women who were overweight. In other words, the effect of aging was stronger than the effect of body size on cholesterol metabolism. The most likely reason for this difference is the effect of menopause. Women who are underweight have a 30% greater risk of early menopause [1]. Cross-sectional data from large-scale population studies suggest that around the time of menopause, low-density lipoprotein (LDL)-cholesterol levels increase by approximately 15 to 25% [25]. Total cholesterol in obese and normal menopausal women was similar at 201 ± 25 and 206 ± 23 mg/dL, although the data from Korea did not include data for underweight individuals [26]. Estrogen or hormone replacement therapy can improve LDL- and HDL-C levels and decrease Lp(a) levels [25]. In Korean women, non-HDL-C levels simultaneously increase at 3.42 mg/dL per year (95% CI, 3.11 to 3.72 mg/dL) during the menopausal transition [27]. In addition, in postmenopausal women, a lower BMI is associated with lower bone mineral density [28], and the low estrogen concentrations that result from menopause accelerate the development of dyslipidemia and atherosclerosis [29,30]. Thus, non-HDL-C and LDL-C are known to increase with menopause, but there have been no reports on whether this phenomenon applies to underweight women. In this study, we found for the first time that total cholesterol and non-HDL-C increased in underweight women as well as in overweight women during the aging and menopause transition.

HDL-C behaves differently from cholesterol and non-HDL-C; HDL-C increases with age, but the difference between overweight and underweight individuals increases inversely. This may appear to be a benefit of being underweight, but this is not the case. In the present study, the HDL-C concentration was greater in underweight women than in overweight women. Several previous studies have shown that higher HDL-C concentrations may not prevent atherosclerosis [31], and no previous clinical trials have evaluated the efficacy of increasing circulating HDL-C concentrations for reducing cardiovascular risk. Therefore, the focus of related research has shifted toward a better understanding of the protective effects of HDL in the vasculature [29,30,31]. Several previous studies have shown that as women progress through menopause, an increase in the HDL-C concentration is independently associated with an increase in carotid intima-media thickness [31]. Therefore, in underweight women aged 50–65 years, high HDL-C concentrations do not necessarily protect against atherosclerosis. However, whether HDL function (cholesterol efflux capacity), estrogen concentration, and cardiovascular risk differ between this population and women aged >20–39 years or between women with obesity of the same age requires investigation.

In addition to aging, HbA1c and TG are more strongly influenced by body size and HDLc. It is well known that these factors are strongly influenced by insulin sensitivity; it is also well known that HDL-C and TG move in opposite directions as insulin sensitivity decreases. TG and HbA1c levels are significantly affected by insulin. Insulin secretion decreases with age, insulin resistance increases with weight gain, and TG and HbA1c levels exhibited corresponding differences according to age and body size in the present study. Furthermore, the differences between underweight and overweight women increased with age and were greatest in the 50–65-year-old group. There was no difference in HbA1c between women who were underweight or had a normal weight, but the difference in TG concentration between those who were underweight and those with a normal weight increased with age. In contrast, the differences in HbA1c between the underweight and normal-weight groups were not significant, even for women aged 50–65 years. It has previously been reported that young women who are underweight have a high incidence of impaired glucose tolerance [32] and that postmenopausal women who are underweight are more likely to have impaired glucose tolerance than young women who are underweight [32]. The authors of this study also found that the HbA1c levels of underweight women were similar to those of women with a normal weight [32]. As in these papers, differences can be detected in glucose loading, such as OGTT results; however, clinically, it may be difficult to observe differences in HbA1c in underweight individuals because they eat less than individuals with a normal weight. Therefore, the results of these previous studies are consistent with those of the present study. Taken together, these findings suggest that even small differences in body mass, such as between individuals with relatively low and normal body masses, can affect insulin sensitivity. Since the difference between underweight and normal weight with respect to HbA1c and TG levels is clinically acceptable, it would be preferable to have control at a normal weight rather than underweight in view of other health risks.

The limitation of the present study was the lack of information regarding the menstrual cycle and lifestyle (exercise, smoking, alcohol, and comorbidities) of the participants. Furthermore, the effect of menopause could not be evaluated. The mean age at menopause in Japanese women is 50.5 years, and the 5 years before and after menopause are referred to as the pre- and postmenopausal stages, respectively [33,34]. Therefore, we assumed that most of the participants aged 50–65 years were perimenopausal. Thus, additional detailed information should be collected in the future to evaluate the effect of menopause on circulating lipid concentrations. Moreover, HbA1c may be affected by anemia, pregnancy, abnormal hemoglobin levels, and steroid treatment [35,36,37]. HbA1c reflects glucose tolerance better than occasional blood glucose measurements at the population level. Blood glucose concentrations, including those associated with oral glucose tolerance testing, are required to assess an individual’s glucose tolerance. Information about the lifestyle (exercise, smoking, alcohol consumption, and comorbidities) of the participants is important for estimating cardiovascular risk [38]. Because we obtained and analyzed processed information that excluded personal information after the checkup, we unfortunately do not have these data in this study. Finally, because this was a cross-sectional study rather than a longitudinal or cohort study of year-to-year changes in individuals, we could evaluate trends only in the sample from year to year. Therefore, we would like to conduct a retrospective case-control study in collaboration with several screening centers in the future.

## 5. Conclusions

In conclusion, the proportion of women who were underweight decreased with age but remained at 10% for those aged 50–65 years. TC and non-HDL-C concentrations increased with age, regardless of body size, and these concentrations in women who were underweight and aged 50–65 years were comparable to those in women who were overweight and of the same age. Although an increase in cholesterol with age has been previously reported, this was the first study to show that this change occurred regardless of body size. Since cholesterol levels are strongly influenced by aging and less by body size, it appeared that even underweight people need to limit saturated fatty acid intake and consume dietary fiber, suggesting that dietary habits such as reducing the intake of processed meat and consuming fiber are necessary even at a young age. On the other hand, HDL-C, TG, and HbA1c, which are influenced by insulin sensitivity, were strongly affected by body size with age. Results for TG, HDL-C, and HbA1c showed no significant difference between normal and underweight, suggesting that normal weight should be maintained for risk of disease. 

These results suggested that instead of forcing underweight women to consume a diet that forces them to gain weight, it is necessary to encourage them to improve their lifestyle by consuming foods that contain vitamins and trace elements, which are often lacking, and by missing breakfast. In the future, it is necessary to collect information on daily food intake and lifestyle history during staff health checkups to improve the living environment of staff.

## Figures and Tables

**Table 1 healthcare-12-00465-t001:** Baseline characteristics of the participants at their health examinations between 2012 and 2022.

Total	Total (n = 26,118)	Underweight (n = 3641, 14%)	Normal Weight (n = 19,895, 76%)	Overweight (n = 2582, 10%)
Age	34.3 (10.8)	31.4 (9.4)	34.1 (10.8)	39.7 (11.1)
BMI	21.2 (3.1)	17.6 (0.7)	21.0 (1.6)	27.9 (3.0)
TC	189.7 (32.8)	181.0 (29.6)	189.5 (32.6)	203.7 (34.0)
TG	87.4 (58.4)	71.0 (35.9)	84.4 (54.0)	133.8 (87.7)
HDLC	71.0 (14.6)	74.8 (14.6)	71.6 (14.1)	61.3 (14.0)
Non-HDL-C	118.7 (31.4)	106.3 (25.1)	117.9 (30.4)	142.4 (33.8)
HbA1c	5.42 (0.36)	5.37 (0.22)	5.39 (0.28)	5.71 (0.76)
**20–29 y.o.**	**Total (n = 11,840)**	**Underweight (n = 2059, 17%)**	**Normal Weight (n = 9171, 78%)**	**Overweight (n = 610, 5%)**
Age	24.8 (2.3)	24.9 (2.2)	24.7 (2.3)	25.0 (2.4)
BMI	20.5 (2.5)	17.6 (0.7)	20.7 (1.5)	27.7 (2.7)
TC	179.9 (28.8)	174.5 (25.1)	180.5 (29.2)	190.0 (31.2)
TG	73.7 (42.7)	66.5 (31.2)	73.0 (41.1)	108.3 (73.1)
HDLC	70.4 (13.5)	72.6 (12.6)	70.7 (13.4)	60.0 (13.9)
Non-HDL-C	109.5 (26.4)	101.9 (21.9)	109.8 (26.2)	130.0 (31.2)
HbA1c	5.33 (0.25)	5.32 (0.20)	5.33 (0.24)	5.44 (0.46)
**30–39 y.o.**	**Total (n = 6160)**	**Underweight (n = 882, 14%)**	**Normal Weight (n = 4666, 76%)**	**Overweight (n = 612, 10%)**
Age	34.0 (2.9)	33.7 (2.8)	34.0 (2.9)	34.4 (2.9)
BMI	21.2 (3.1)	17.7 (0.7)	20.9 (1.6)	28.1 (3.3)
TC	186.6 (31.1)	180.7 (24.5)	186.0 (30.9)	199.7 (34.9)
TG	87.6 (54.0)	70.6 (33.6)	85.1 (50.2)	131.2 (78.3)
HDLC	70.8 (14.2)	74.9 (13.9)	71.3 (13.7)	60.9 (14.3)
Non-HDL-C	115.8 (29.9)	105.7 (23.1)	114.7 (29.1)	138.9 (32.3)
HbA1c	5.39 (0.33)	5.38 (0.21)	5.37 (0.23)	5.60 (0.77)
**40–49 y.o.**	**Total (n = 5019)**	**Underweight (n = 449, 9%)**	**Normal Weight (n = 3765, 75%)**	**Overweight (n = 805, 16%)**
Age	44.4 (2.8)	44.2 (2.9)	44.4 (2.8)	44.7 (2.8)
BMI	22.0 (3.4)	17.7 (0.7)	21.2 (1.6)	28.1 (3.1)
TC	198.1 (30.8)	190.7 (32.2)	196.7 (30.0)	208.8 (31.3)
TG	98.7 (66.9)	81.8 (46.9)	92.2 (60.2)	138.2 (88.7)
HDLC	71.8 (15.8)	78.7 (17.5)	72.9 (15.2)	62.5 (13.8)
Non-HDL-C	126.3 (30.7)	112.0 (26.9)	123.8 (28.8)	146.4 (32.2)
HbA1c	5.49 (0.43)	5.47 (0.25)	5.43 (0.27)	5.78 (0.839
**50–65 y.o.**	**Total (n = 3099)**	**Underweight (n = 251, 8%)**	**Normal Weight (n = 2293, 74%)**	**Overweight (n = 555, 18%)**
Age	54.6 (3.7)	54.5 (3.6)	54.6 (3.7)	54.6 (3.9)
BMI	22.4 (3.4)	17.5 (0.8)	21.6 (1.71)	27.8 (3.1)
TC	219.5 (32.6)	218.9 (35.6)	220.5 (31.9)	215.5 (33.9)
TG	121.2 (81.8)	90.6 (45.0)	115.5 (76.4)	158.1 (102.2)
HDLC	72.2 (16.9)	85.3 (19.8)	73.4 (15.8)	61.4 (14.0)
Non-HDL-C	147.2 (32.7)	133.6 (32.6)	147.1 (31.8)	154.1 (34.9)
HbA1c	5.68 (0.49)	5.54 (0.30)	5.62 (0.36)	6.02 (0.79)

**Table 2 healthcare-12-00465-t002:** Comparisons of the mean differences in total cholesterol concentration according to BMI category.

Age	BMI Adjusted Mean (SE)	BMI Adjusted Mean (SE)	Mean Difference [95%CI]	*p*
20–29 y.o. (n = 11,840)	underweight (n = 2059)	normal weight (n = 9171)	−6.0 [−7.4, −4.6]	<0.001 *
	174.5 (0.66)	180.5 (0.3)		
		Overweight (n = 610)	−15.5 [−18.2, −12.8]	<0.001 *
		190.0 (1.2)		
30–39 y.o. (n = 6160)	underweight (n = 882)	normal weight (n = 4666)	−5.3 [−7.5, −3.2]	<0.001 *
	180.8 (1.0)	186.1 (0.4)		
		overweight (n = 612)	−18.9 [−22.0, −15.8]	<0.001 *
		199.7 (1.2)		
40–49 y.o. (n = 5019)	underweight (n = 449)	normal weight (n = 3765)	−6.0 [−8.9, −3.1]	<0.001 *
	190.8 (1.4)	196.8 (0.5)		
		overweight (n = 805)	−18.1 [−21.5, −14.6]	<0.001 *
		208.8 (1.1)		
50–65 y.o. (n = 3099)	underweight (n = 251)	normal weight (n = 2293)	−1.8 [−5.7, 2.1]	1
	218.5 (1.9)	220.3 (0.6)		
		overweight (n = 555)	3.4 [−1.0, 7.9]	1
		215.1 (1.3)		

The means were adjusted for the year 2017 and categorized by age and BMI. * *p* < 0.001, *p* values adjusted for the Bonferroni method.

**Table 3 healthcare-12-00465-t003:** Comparisons of the mean differences in non-HDL-C concentrations according to BMI category.

Age	BMI Adjusted Mean (SE)	BMI Adjusted Mean (SE)	Mean Difference [95%CI]	*p*
20–29 y.o. (n = 11,840)	underweight (n = 2059)	normal weight (n = 9171)	−7.8 [−9.2, −6.5]	<0.001 *
	102.0 (0.6)	109.8 (0.3)		
		overweight (n = 610)	−27.9 [−30.4, −25.4]	<0.001 *
		129.9 (1.1)		
30–39 y.o. (n = 6160)	underweight (n = 882)	normal weight (n = 4666)	−8.9 [−10.9, −6.9]	<0.001 *
	106.1 (0.9)	115.0 (0.4)		
		overweight (n = 612)	−32.7 [−35.6, −29.8]	<0.001 *
		138.8 (1.1)		
40–49 y.o. (n = 5019)	underweight (n = 449)	normal weight (n = 3765)	−11.7 [−14.4, −9.0]	<0.001 *
	112.3 (1.3)	124.0 (0.5)		
		overweight (n = 805)	−34.1 [−37.3, −30.9]	<0.001 *
		146.4 (1.0)		
50–65 y.o. (n = 3099)	underweight (n = 251)	normal weight (n = 2293)	−13.9 [−17.5, −10.3]	<0.001 *
	132.6 (1.8)	146.4 (0.6)		
		overweight (n = 555) 152.8 (1.2)	−20.3 [−24.4, −16.1]	<0.001 *

The means were adjusted for the year 2017 and categorized by age and BMI category. * *p* < 0.001, *p* values adjusted for the Bonferroni method.

**Table 4 healthcare-12-00465-t004:** Comparisons of the mean differences in the HDL-C concentration according to BMI category.

Age	BMI Adjusted Mean (SE)	BMI Adjusted Mean (SE)	Mean Difference [95%CI]	*p*
20–29 y.o. (n = 11,840)	underweight (n = 2059)	normal weight (n = 9171)	1.8 [1.2, 2.5]	<0.001 *
	72.5 (0.3)	70.7 (0.2)		
		overweight (n = 610)	12.5 [11.2, 13.7]	<0.001 *
		60.0 (0.6)		
30–39 y.o. (n = 6160)	underweight (n = 882)	normal weight (n = 4666)	3.6 [2.6, 4.6]	<0.001 *
	74.7 (0.5)	71.1 (0.2)		
		overweight (n = 612)	13.8 [12.4, 15.2]	<0.001 *
		60.9 (0.6)		
40–49 y.o. (n = 5019)	underweight (n = 449)	normal weight (n = 3765)	5.7 [4.4, 7.1]	<0.001 *
	78.5 (0.7)	72.8 (0.2)		
		overweight (n = 805)	16.0 [14.4, 17.6]	<0.001 *
		62.5 (0.5)		
50–65 y.o. (n = 3099)	underweight (n = 251)	normal weight (n = 2293)	12.1 [10.3, 13.9]	<0.001 *
	85.9 (0.9)	73.8 (0.3)		
		overweight (n = 555)	23.7 [21.6, 25.7]	<0.001 *
		62.3 (0.6)		

The means were adjusted for the year 2017 and categorized by age and BMI. * *p* < 0.001, *p* values adjusted for the Bonferroni method.

**Table 5 healthcare-12-00465-t005:** Comparisons of the mean differences in triglyceride concentrations according to BMI category.

Age	BMI Adjusted Mean (SE)	BMI Adjusted Mean (SE)	Mean Difference [95%CI]	*p*
20–29 y.o. (n = 11,840)	underweight (n = 2059)	normal weight (n = 9171)	−6.5 [−9.1, −3.9]	<0.001 *
	66.5 (1.2)	73.0 (0.6)		
		overweight (n = 610)	−41.8 [−46.7, −36.9]	<0.001 *
		108.3 (2.2)		
30–39 y.o. (n = 6160)	underweight (n = 882)	normal weight (n = 4666)	−14.5 [−18.4, −10.6]	<0.001 *
	70.6 (1.8)	85.1 (0.8)		
		overweight (n = 612)	−60.7 [−66.3, −55.0]	<0.001 *
		131.2 (2.2)		
40–49 y.o. (n = 5019)	underweight (n = 449)	normal weight (n = 3765)	−10.4 [−15.7, −5.0]	<0.001 *
	81.8 (2.6)	92.2 (0.9)		
		overweight (n = 805)	−56.3 [−62.6, −50.0]	<0.001 *
		138.2 (1.9)		
50–65 y.o. (n = 3099)	underweight (n = 251)	normal weight (n = 2293)	−24.9 [−32.1, −17.8]	<0.001 *
	90.6 (3.4)	115.5 (1.1)		
		overweight (n = 555)	−67.5 [−75.7, −59.4]	<0.001 *
		158.1 (2.3)		

The means were adjusted for the year 2017 and categorized by age and BMI. * *p* < 0.001, *p* values adjusted for the Bonferroni method.

**Table 6 healthcare-12-00465-t006:** Comparisons of the mean differences in HbA1c according to BMI category.

Age	BMI Adjusted Mean (SE)	BMI Adjusted Mean (SE)	Mean Difference [95%CI]	*P*
20–29 y.o. (n = 11840)	underweight (n = 2059)	normal weight (n = 9171)	−0.006 [−0.022, 0.010]	1
	5.32 (0.007)	5.33 (0.003)		
		overweight (n = 610)	−0.12 [−0.15, −0.087]	<0.001 *
		5.44 (0.014)		
30–39 y.o. (n = 6160)	underweight (n = 882)	normal weight (n = 4666)	0.008 [−0.016, 0.032]	1
	5.38 (0.011)	5.37 (0.005)		
		overweight (n = 612)	−0.23 [−0.27, −0.18]	<0.001 *
		5.60 (0.014)		
40–49 y.o. (n = 5019)	underweight (n = 449)	normal weight (n = 3765)	0.036 [0.003, 0.069]	0.49
	5.47 (0.016)	5.43 (0.005)		
		overweight (n = 805)	−0.31 [−0.35, −0.27]	<0.001 *
		5.78 (0.012)		
50–65 y.o. (n = 3099)	underweight (n = 251)	normal weight (n = 2293)	−0.072 [−0.12, −0.028]	0.099
	5.54 (0.021)	5.62 (0.007)		
		overweight (n = 555)	−0.48 [−0.53, −0.43]	<0.001 *
		6.02 (0.014)		

The adjusted mean was controlled for the year 2017 and categorized by age and BMI. * *p* < 0.001, *p* values adjusted for the Bonferroni method.

## Data Availability

Some or all datasets generated during and/or analyzed during the current study are not publicly available but are available from the corresponding author upon reasonable request.

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
