# Peer review of "Differing Effects of Body Size on Circulating Lipid Concentrations and Hemoglobin A1c Levels in Young and Middle-Aged Japanese Women"

_healthcare, 2024, doi:10.3390/healthcare12040465_

Round 1

Reviewer 1 Report

Comments and Suggestions for Authors

The abstract is unclear and does not describe the essence of the issue. It is necessary to revise it.

Blood samples were collected at approximately 16:00–17:00 from non-fasting participants. Why was this methodology chosen and the samples were not taken in the fasting state?

I do not understand the dispersion of the representation of women in individual categories. How can there be 20-25% women in the 30-39 age group? In the table, you indicate the exact number - a specific number. Unify it. I understand that over time the numbers change, but methodologically this is inappropriate. At the end of the study, you differentiated the women into individual groups.

Did you also find out the anamnesis of diseases in women? Did the women have anorexia or bulimia?

A more detailed and adequate discussion and comparison of own results with the results of other authors is missing, or deeper analysis of the obtained results.

Like the abstract, the conclusion is too brief and inconclusive.

Author Response

#Reviewer 1

Q1: The abstract is unclear and does not describe the essence of the issue. It is necessary to revise it.

Thank you for your suggestions. We rewrote our manuscript according to your suggestions.

Q2: Blood samples were collected at approximately 16:00–17:00 from non-fasting participants. Why was this methodology chosen and the samples were not taken in the fasting state?

A: As the reviewer pointed out, the blood was not drawn on an empty stomach. Since this is a hospital staff medical checkup, the checkup is set for after 3:30 p.m., when staff members have finished seeing outpatients. This may be due in part to the time of day when staff members are more likely to be available for medical checkups. It is not based on scientific evidence, but on the convenience of the hospital.

Line 94-95

Since this is a hospital staff medical checkup, the checkup is set for after 3:30 p.m., when medical staffs have finished seeing outpatients.

Q3: I do not understand the dispersion of the representation of women in individual categories. How can there be 20-25% women in the 30-39 age group? In the table, you indicate the exact number - a specific number. Unify it. I understand that over time the numbers change, but methodologically this is inappropriate. At the end of the study, you differentiated the women into individual groups.

A: According to your suggestions, we unified both the proportion (%) and numbers.

Line113-123

Forty-five percent of the participants (n=11840) were aged >20–29 years, 24% (n=6160) were aged 30–39 years, 19% (n=5019) were aged 40–49 years, and 12% (n=3099) were older than 50 years (Table 1). We identified the metabolic indices of the participants who were underweight, normal weight, or overweight in each age group. The proportions of participants in the 20- to 29-year-old group, 30- to 39-year-old group, 40- to 49-year-old group, and 50- to 65-year-old group were 45% (n=11840), 24% (n=6160), 19% (n=5019), and 12% (n=3099), respectively. The 20–29-year-old group had the highest proportion of participants who were underweight (17%)  and the proportion decreased across the 30–39 (14%), 40–49 (8.9%), and 50–65-year-old age groups (8.1%). However, the 20–29-year-old group had the lowest proportion of participants (5%) and the proportion increased to 18% in the 50–65-year-old age group (Table 1).

Q:Did you also find out the anamnesis of diseases in women? Did the women have anorexia or bulimia?

A: We have no more data on 10 years of screening data than what we have analyzed here. However, the age group for anorexia nervosa is a bit younger (adolescents) and is likely to be less common. Some patients (BMI 17.5) were actually seen as outpatients, but screening (EAT-26>20), none of them were positive.

Q4: A more detailed and adequate discussion and comparison of own results with the results of other authors is missing, or deeper analysis of the obtained results.

Like the abstract, the conclusion is too brief and inconclusive.

A: Thank you for your suggestions. We rewrote them.

Reviewer 2 Report

Comments and Suggestions for Authors

The manuscript by Katsumi Iizuka et al. explores the effects of body size on serum lipid and hemoglobin A1c levels of Japanese women in a cross-sectional study. This work focuses on an interesting area; however, I do have several concerns. Please, see my comments and suggestions below.

1. Although it is a study of interest and that it is needed, the study lacks a methodological quality with the control and identification of biases, as well as the design of the study and all its methodology. In the Materials and Methods section, please add criteria that are including and excluding subjects from the study population. Additionally, add a new Figure - a flow chart of the selection of study participants.

2. The statistical analyses need to take into account other (major) confounding variables such as the presence of any clinical condition or medication/supplementation use as well as smoking status, drinking status, and physical activity level.

3. General characteristics of the patients are insufficient. Data on medical history must be included and also lifestyle parameters and physical activity level of the participants included in this study.

4. Please provide a more detailed discussion of the differences in these blood parameters that occurred between 2012 and 2022 in the four age groups. There is a lack of trend analysis. What factors influence the observed differences presented in figures 2, 3, 4, and 5? What is the purpose of such a comparison if the study does not include analysis in a repeated measures design? 

5. Please adjust Results and Discussion section and the calculations in line with the suggestions for the Materials and Methods section.

6. At the end of the section Discussion, please highlight to a greater extent the following themes: implications of this study and novelty in this study.

7. After implementing the above corrections please verify the abstract as well as make corrections to the relevant sections.

Comments on the Quality of English Language

Minor editing of English language is required. 

Author Response

#Reviewer2

The manuscript by Katsumi Iizuka et al. explores the effects of body size on serum lipid and hemoglobin A1c levels of Japanese women in a cross-sectional study. This work focuses on an interesting area; however, I do have several concerns. Please, see my comments and suggestions below.

Q1: Although it is a study of interest and that it is needed, the study lacks a methodological quality with the control and identification of biases, as well as the design of the study and all its methodology. In the Materials and Methods section, please add criteria that are including and excluding subjects from the study population. Additionally, add a new Figure - a flow chart of the selection of study participants.

A1: Thank you very much for your comments. The data are from medical examinations and there are no exclusion criteria. The point to be laid out is that only female data were used.

Q2: The statistical analyses need to take into account other (major) confounding variables such as the presence of any clinical condition or medication/supplementation use as well as smoking status, drinking status, and physical activity level. 

A2: The reviewer's point is very important. The information we had available was the only information analyzed in this study and did not include information on hospital visits, smoking, alcohol consumption, or physical activity. Therefore, we have stated this point in the limitations of the study.

Line 306

The limitation of the present study was the lack of information regarding the menstrual cycle and lifestyle (exercise, smoking, alcohol, and comorbidities) of the participants. Furthermore, the effect of menopause could not be evaluated. The mean age at menopause in Japanese women is 50.5 years, and the 5 years before and after menopause are referred to as the pre- and postmenopausal stages, respectively [34,35]. Therefore, we assumed that most of the participants aged 50–65 years were perimenopausal. Thus, additional detailed information should be collected in the future to evaluate the effect of menopause on circulating lipid concentrations. Moreover, HbA1c may be affected by anemia, pregnancy, abnormal hemoglobin levels, and steroid treatment [36-38]. HbA1c reflects glucose tolerance better than occasional blood glucose measurements at the population level. Blood glucose concentrations, including those associated with oral glucose tolerance testing, are required to assess an individual’s glucose tolerance. Information about the lifestyle (exercise, smoking, alcohol consumption, and comorbidities) of the participants is important for estimating cardiovascular risk [39]. Because we obtained and analyzed processed information that excluded personal information after the checkup, we unfortunately do not have these data in this study. Finally, because this was a cross-sectional study rather than a longitudinal or cohort study of year-to-year changes in individuals, we could evaluate trends only in the sample from year to year. Therefore, we would like to conduct a retrospective case‒control study in collaboration with several screening centers in the future.

Q3. General characteristics of the patients are insufficient. Data on medical history must be included and also lifestyle parameters and physical activity level of the participants included in this study.

A3: Unfortunately, since the data are from staff medical examinations, details of treatment history data are not available.

Q4: Please provide a more detailed discussion of the differences in these blood parameters that occurred between 2012 and 2022 in the four age groups. There is a lack of trend analysis. What factors influence the observed differences presented in figures 2, 3, 4, and 5? What is the purpose of such a comparison if the study does not include analysis in a repeated measures design?  

A4: According to your suggestions, we omitted Figure 2-6.

Q5. Please adjust Results and Discussion section and the calculations in line with the suggestions for the Materials and Methods section.

A5: According to your suggestions, we thoroughly rewrote it.

Q6. At the end of the section Discussion, please highlight to a greater extent the following themes: implications of this study and novelty in this study.

Aï¼–: We are the first in the world to show that underweight women have increased cholesterol and non-HDLC as well as normal weight and overweight women.

Q7. After implementing the above corrections please verify the abstract as well as make corrections to the relevant sections.

Aï¼—; Thank you for your comments. We corrected them.

Round 2

Reviewer 2 Report

Comments and Suggestions for Authors

The authors have addressed the questions raised in the first round of the review. They have corrected the text according to my suggestions. I only have one minor comment: on pages 3-4, Table 1, for each age group in the header, please verify the numerical values expressed as (n=….); the percentages appear to be incorrectly listed. 

Author Response

Thank you for pointing this out.
We have checked and found no errors in the ratios and numbers. It is difficult to understand, so I have listed both the number and the ratio together. The leftmost total is the total number. The total on the far left is the total number, and the numbers and ratios of underweight, normal weight, and excess weight to that number. Upon review, some errors were found in some of the tables and have been corrected (Table 4).